# Gene Expression Analysis of Yeast Strains with a Nonsense Mutation in the eRF3-Coding Gene Highlights Possible Mechanisms of Adaptation

**DOI:** 10.3390/ijms25126308

**Published:** 2024-06-07

**Authors:** Evgeniia M. Maksiutenko, Yury A. Barbitoff, Lavrentii G. Danilov, Andrew G. Matveenko, Olga M. Zemlyanko, Elena P. Efremova, Svetlana E. Moskalenko, Galina A. Zhouravleva

**Affiliations:** 1Department of Genetics and Biotechnology, St. Petersburg State University, 199034 St. Petersburg, Russia; jmrose@yandex.ru (E.M.M.); barbitoff@bioinf.me (Y.A.B.); l.danilov@spbu.ru (L.G.D.); a.matveenko@spbu.ru (A.G.M.); o.zemlyanko@spbu.ru (O.M.Z.); efremova.bio@gmail.com (E.P.E.); smoskalenko@mail.ru (S.E.M.); 2St. Petersburg Branch, Vavilov Institute of General Genetics of the Russian Academy of Sciences, 199034 St. Petersburg, Russia; 3Bioinformatics Institute, 197342 St. Petersburg, Russia; 4Laboratory of Amyloid Biology, St. Petersburg State University, 199034 St. Petersburg, Russia

**Keywords:** translation termination, release factors, yeast, eRF3, nonsense mutations, cell cycle, RNA-seq, proteome, *CDC20*, *CDC23*, *CDC28*, *FKH1*

## Abstract

In yeast *Saccharomyces cerevisiae*, there are two translation termination factors, eRF1 (Sup45) and eRF3 (Sup35), which are essential for viability. Previous studies have revealed that presence of nonsense mutations in these genes leads to amplification of mutant alleles (*sup35-n* and *sup45-n*), which appears to be necessary for the viability of such cells. However, the mechanism of this phenomenon remained unclear. In this study, we used RNA-Seq and proteome analysis to reveal the complete set of gene expression changes that occur during cellular adaptation to the introduction of the *sup35-218* nonsense allele. Our analysis demonstrated significant changes in the transcription of genes that control the cell cycle: decreases in the expression of genes of the anaphase promoting complex APC/C (*APC9*, *CDC23*) and their activator *CDC20*, and increases in the expression of the transcription factor *FKH1*, the main cell cycle kinase *CDC28*, and cyclins that induce DNA biosynthesis. We propose a model according to which yeast adaptation to nonsense mutations in the translation termination factor genes occurs as a result of a delayed cell cycle progression beyond the G2-M stage, which leads to an extension of the S and G2 phases and an increase in the number of copies of the mutant *sup35-n* allele.

## 1. Introduction

Termination of protein synthesis occurs when the release factor recognizes a stop codon (UAA, UAG, or UGA) and stimulates a nascent peptide release (reviewed in [1]). About 11% of all human disease-associated genetic variants are nonsense mutations, resulting in the occurrence of a premature translation-termination codon (PTC) in the protein-coding gene sequence [2]. Translation of the PTC-containing mRNAs leads to synthesis of truncated, often dysfunctional, polypeptides that can have a dominant-negative or gain-of-function effect on gene function. More and more frequently, therapeutic approaches for disorders caused by nonsense mutations use drugs that promote PTC readthrough, forcing the translational machinery to recode an in-frame PTC into a sense codon in a process called nonsense suppression. Thus, nonsense suppression therapy refers to an effective means of preventing protein translation termination and alleviating disease symptoms by inducing PTC readthrough [3,4].

One of the most popular model objects for studying translation termination and nonsense suppression is the baker’s yeast *Saccharomyces cerevisiae*. In yeast, the mechanisms of adaptation to translation defects could be investigated using strains carrying mutations in one of the genes encoding translation termination factors, *SUP45* (encoding yeast eRF1) [5] and *SUP35* (eRF3) [6,7]. Both of these genes are essential in yeast, and deletion of either one leads to the death of the yeast cells. However, viable strains with nonsense mutations in both *SUP45* [8] and *SUP35* [9] have previously been isolated in our laboratory and extensively characterized.

It has been shown that nonsense mutations (denoted as *sup35-n*) lead to the formation of truncated proteins and a significant (up to 100-fold) decrease in the level of the full-length eRF3 protein. Previously, we demonstrated that the introduction of these *sup35-n* alleles is associated with gene amplification in the corresponding strains [10]. Specifically, the number of copies of the plasmid bearing the mutant alleles of the *SUP35* gene was significantly increased compared to strains bearing plasmids with the wild-type allele. Furthermore, strains carrying the *sup35-218* allele as a single chromosomal copy contained duplication of a part of a chromosome IV (bearing the *SUP35* gene), while strains with other nonsense mutations had additional copies of chromosomes II (*sup35-203*), XI (*sup35-240*), or XIII (*sup35-244*, *sup35-260*).

In this work, we investigated the mechanisms of yeast adaptation to mutations in release factor genes using one of the *sup35-n* alleles, *sup35-218*. The *sup35-218* mutation is located in the N-proximal part of the *SUP35* gene, and leads to the G > T substitution in the 541st nucleotide of the protein-coding sequence, resulting in the emergence of a PTC (TAA) in the 181st position. This leads to the formation of a truncated eRF3 protein that contains 180 amino acids, instead of 685, and completely lacks the functional C-domain. The N- and M-domains that are present in the truncated protein are not directly involved in the process of translation termination, in contrast to the C-domain. Hence, the truncated protein is incapable of stimulation of translation termination. In the *sup35-218* nucleotide sequence, the PTC is located in a “weak” nucleotide context, as the UAA codon is followed by the C nucleotide, which leads to ineffective translation termination at tetranucleotide UAAC [11,12]. Consequently, cells carrying the *sup35-218* allele contain up to 8% of full-length eRF3 due to nonsense suppression [9].

In the present study, we performed a high-throughput transcriptome sequencing (RNA-seq) and proteome analysis to study changes that occur in cells containing the *sup35-218* mutation, and uncover the possible mechanisms of mutant allele amplification. We have identified the genes involved in the cell cycle regulation changes in the level of transcription that may allow cells carrying a plasmid with a nonsense mutant allele to remain viable. Analysis of the RNA-seq and proteome data suggests a plausible molecular mechanism which could explain the basis of cellular adaptation to disturbances in translation termination process.

## 2. Results

### 2.1. Global Transcriptional Changes in Yeast Strains with Nonsense Mutations in the Release Factor Genes

As mentioned earlier, yeast cells are capable of adaptation to nonsense mutations in genes encoding release factors, *SUP35* and *SUP45*. Such an adaptation can be observed when a plasmid carrying the wild-type *SUP35* or *SUP45* allele as a sole source of the corresponding release factor is replaced with another plasmid bearing a mutant allele. To study the mechanisms of yeast adaptation in such a system, cells of the U-14-D1690 strain with a deletion of a normal chromosomal copy of *SUP35* and bearing a wild-type allele of the corresponding gene on a plasmid were transformed with a second plasmid bearing either a wild-type or mutant *sup35-218* allele of the same gene (Figure 1A). Subsequently, strains containing two plasmids were grown on medium containing 5-FOA to lose the plasmid with the *URA3* marker carrying the wild-type *SUP35* allele. Following such plasmid shuffling, we performed a high-throughput transcriptome analysis of the resulting strains (denoted as “3”) and the initial strain (denoted as “1”) using RNA-seq.

Having obtained the RNA-seq data, we first analyzed the total differences in gene expression profiles between samples carrying mutant and wild-type *SUP35* alleles at different stages of the experiment. To do this, we performed principal component analysis (PCA) of the normalized gene count matrix obtained after quantification of gene expression (see Section 4) PCA revealed that the samples perfectly clustered into three groups, corresponding to cells carrying the mutant *sup35-218* alleles or wild-type alleles (at the beginning of the experiment and after the plasmid shuffling) (Figure 1B). A clear clustering of the samples into groups in the space of the first two principal components indicates the presence of reproducible differences in the gene expression profiles in the studied samples. Furthermore, the first of the principal components was sufficient to distinguish between cells bearing the *SUP35* or *sup35-218* allele while explaining as much as 69% of the gene expression variance. This result suggests that introduction of a nonsense allele of *SUP35* has a pronounced effect on the gene expression profile of a yeast cell.

Next, we performed the differential gene expression analysis either at the initial stage (stage 1) of the experiment or after the plasmid shuffling (stage 3). When the gene expression profiles of cells carrying the *sup35-218* allele at stage 3 were compared to those of cells carrying the wild-type *SUP35* allele at stage 1, we discovered 1240 upregulated genes and 1341 downregulated genes for which the adjusted *p*-value was less than 0.05 (Figure 1C). At the same time, analysis of the DEGs identified between cells carrying the mutant allele and cells carrying the wild-type allele after the shuffling reduced the number of genes with increased expression to 1035, and of those with reduced expression to 1100 (Figure 1C). Similar trends were observed when only genes with an absolute log2FC value greater than 0.5 were selected, with a total of 1824 DEGs observed between stages 1 and 3, and 1366 between cells after plasmid shuffling. The number of DEGs was substantially lower if only the genes with a two-fold difference in expression were considered (Figure 1C, right panel); again, more DEGs were observed when the initial strain was used as a control group. A substantial difference in the number of DEGs observed with different sets of control samples suggests that the marker gene used for plasmid maintenance (*LEU2* or *URA3*) also significantly affects the global transcriptional profile of the cell. Given all of the aforementioned observations, we focused our attention on the set of DEGs identified when comparing cells after the plasmid shuffling and with absolute log2FC values greater than 0.5. In this set of genes, 817 genes had an increased expression and 1007 had a decreased expression, and a subset of 147 and 216 genes had at least two-fold increased and decreased expressions, respectively (Figure 1D). Importantly, the set of genes with the most substantial increase in expression included *SUP35* itself, as well as *LEU2*. These observations are in perfect concordance with the results of our earlier work, which showed an increase in the number of plasmid copies in cells bearing *sup35-218* accompanied by significantly elevated expression of this allele detected using qPCR [10].

### 2.2. Identification of Biological Processes Involved in Adaptation to the Presence of sup35-218

To gain insights into the biological mechanisms driving the adaptation to the presence of nonsense mutations in *SUP35*, as well as amplification of the mutant allele, we performed further functional analysis of the obtained list of DEGs. As the first step of such an analysis, we performed the Gene Ontology (GO) enrichment analysis for lists of up- and downregulated DEGs separately. Among the GO biological processes, we discovered enrichment of upregulated DEGs with genes involved in the control of the cell cycle and carbohydrate metabolism (including glucose import) (Figure 2A).

In concordance with these results, a significant overrepresentation of proteins active in the cell wall or as components of cyclin-dependent protein kinase complexes was detected when testing the cellular component enrichment (Figure 2B). Among the genes with reduced expression, overrepresentation of the genes involved in various biosynthesis processes, including the synthesis of amino acids, was found (Figure 2C). While the repression of biosynthetic processes coupled with activation of catabolism and the import of nutrients is expected during stress conditions, the upregulation of the cell cycle genes seemed a more interesting phenomenon. Hence, we next went on to investigate the changes in the yeast cell cycle in more detail.

To perform such an in-depth analysis of the effects of DEGs on the cell cycle, we used the Kyoto Encyclopedia of Genes and Genomes (KEGG, https://www.genome.jp/kegg/kegg2.html, accessed on 10 October 2023), which could be utilized to visualize the position and interaction of the studied genes in the entire pathway diagram. Visualization of the DEGs on a diagram of the yeast cell cycle regulation pathway revealed several interesting patterns (Figure 3). First of all, we noted an increase in the levels of transcription of *CDC28*, the most important regulator of the cell cycle, the transcription cofactor *SWI6*, as well as *MCM2* and *ORC2* genes involved in DNA replication. Secondly, increased expression of genes encoding cyclins of the CLN, CLB family and subunits of the cohesin complex *SMC3* and *MCD1* was detected. Finally, besides an increase in the expression of genes primarily involved in DNA synthesis, a decrease in the transcription of genes involved in the metaphase–anaphase transition (*CDC23*, *APC9*) was noted.

Taken together, these observations suggest that the transition of the yeast cell to mitosis is impaired upon introduction of the nonsense mutant allele of *SUP35*. At the same time, proteins involved in cell cycle progression at earlier stages, as well as in the DNA synthesis, are more active in cells bearing the *sup35-218* allele compared to control cells. These results may provide a mechanistic explanation of the observed phenomena of adaptation (see Section 3).

In addition to the analysis of functions of the identified DEGs, the large number of DEGs suggest that certain transcription factors (TFs) should play a key role in driving the observed global changes in the gene expression profile. To identify such TFs, we used the YEASTRACT (www.yeastract.com, accessed on 20 November 2023) tool.

Our analysis identified the enrichment of target genes of TFs associated with the biosynthesis of amino acids and nucleotides among genes with reduced expression (Appendix A) in cells bearing *sup35-218*. Specifically, overrepresentation of targets was observed for Bas1, which regulates genes of the purine and histidine biosynthesis pathways (55.63% of genes with reduced expression; *p*-value < 10^−15^), and for Gcn4, a transcriptional activator of amino acid biosynthetic genes (93.52% of genes in the studied gene set; *p*-value < 10^−15^). These observations are in good concordance with the results obtained using GO term enrichment analysis.

For the upregulated DEGs (Appendix A), we identified an enrichment of targets of transcription factors involved in metabolism (Ino2, Gcn4) and stress response (Rpn4 and Hsf). Moreover, we discover that a significant number of the identified DEGs are targets of Rph1, Mig1, and Ste12, which are involved in many biological processes such as transcription, autophagy, invasive growth, and others.

Interestingly, an overrepresentation of the targets of several transcription factors involved in cell cycle regulation was found (specifically, Aft1, Ybp1, Mcm1, and Ume6). It was shown that Aft1, a protein which regulates iron homeostasis, has a role in chromosome stability, interaction with the kinetochore and promoting pericentric cohesin [13]. Yhp1 is a homeobox transcriptional repressor; it binds Mcm1 and early cell cycle box (ECB) elements of cell cycle regulated genes, thereby restricting ECB-mediated transcription to the M/G1 interval [14]. Ume6 is a histone deacetylase complex subunit and a key transcriptional regulator of early meiotic genes, which are involved in chromatin remodeling and transcriptional repression [15,16].

### 2.3. Validation of High-Throughput Sequencing Results Using qPCR

Given that the cell cycle progression is a finely regulated process, many of the identified DEGs corresponding to cell cycle regulators have relatively small changes in expression (1.4- to 2-fold, corresponding to log2FC values between 0.5 and 1). This prompted us to conduct additional validation of discovered changes in expression with the qPCR method for the most significant cell cycle genes. In total, six genes were selected for validation, including three of the upregulated ones (*FKH1*, *CDC28*, and *SUP35*), two of the downregulated ones (*CDC20* and *CDC23*), and one gene with no changes in expression according to RNA-seq data (*SUP45*). Two of the four cell cycle genes that were chosen for validation had an absolute value of log2FC <1 (*CDC20* and *CDC28*).

For all of the genes selected for validation, the observed difference in median expression from qPCR corroborated transcriptomic analysis, and statistically significant differences in expression were confirmed for four out of five DEGs (Figure 4). Thus, we confirmed a decrease in the expression of the *CDC20* gene in cells bearing the *sup35-218* allele. For another gene involved in the anaphase transition, *CDC23*, the difference in median expression was not statistically significant. In good concordance with RNA-seq data, median expression of *CDC28*, encoding the main kinase that regulates the cell cycle in yeast, was twice higher in the presence of a nonsense allele compared to cells containing the wild-type *SUP35* gene. For the *FKH1* transcription factor involved in the regulation of cyclin levels, we observed a significant increase in relative expression levels, which is also in line with the RNA-seq analysis. The *SUP35* gene normally has increased mRNA levels in the cells harboring the *sup35-218* allele, also demonstrating a solid upregulation in qPCR. Finally, the *SUP45* gene, which was not differentially expressed according to both RNA-seq data and our previous results, demonstrated no significant expression changes in qPCR validation.

Thus, we confirmed that, in the presence of the *sup35-218* mutant allele in yeast cells, an increase in the transcription of the *CDC28* and *FKH1* genes and a decrease in the expression of the activator of anaphase-promoting complex *CDC20* occurred. We suggest that these changes can affect the duration of cell cycle phases.

### 2.4. Proteome Analysis of Cells Harboring sup35-218 Allele

To better understand the changes that occur because of disturbances in the translation termination process, we also evaluated changes in cells at the protein level. For this purpose, we performed proteome analysis of the same cells harboring *sup35-218* mutation. By examining the proteome, we identified 117 differentially produced proteins (56 with increased production and 62 with decreased production) (Figure 5A). However, our analysis of GO terms for differentially produced proteins showed no enrichment.

We also assessed the similarity of the observed changes in the proteome and transcriptome (Appendix A). For this purpose, we evaluated the correlation between the protein and transcript levels for each gene. We have detected a statistically significant moderate correlation (Figure 5B) between the transcript levels and protein levels for both wild-type strains and strains with the *sup35-218* mutation (correlation between proteome and transcriptome in wild-type strain–tau = 0.256, *p*-value <0.001; correlation between proteome and transcriptome in strain with *sup35-218* mutation–tau = 0.238, *p*-value <0.001). While a low degree of correspondence between the protein and transcript levels has previously been reported for yeast [17,18,19], a lack of differences in the correlation coefficients suggests that nonsense mutations in *SUP35* do not disrupt the regulation of gene expression at translational level.

Although a poor correlation was observed between the transcriptome and proteome, there were still several genes whose changes in expression levels are consistent with their protein levels. Genes *RPC82*, *STU2*, *CDC28*, *PST1* and their protein products were upregulated in transcriptome and proteome. At the same time, expression and protein levels of *FOL2*, *ERG26*, *ERG28*, *HMF1*, *BNA1* were decreased (Table 1). Despite the low number of overlaps between the datasets, the functions of these nine proteins further support the conclusions made based on functional annotation of RNA-Seq results. Particularly, two of the upregulated proteins control cell cycle and division (Cdc28 and Stu2); moreover, an increased abundance of Rpc82, which is responsible for tRNA biosynthesis, is consistent with transcriptional downregulation of amino acid biosynthesis genes.

### 2.5. Effects of the sup35-218 Mutation on Cell Cycle Progression

Since the proteomic and transcriptomic analyses revealed several overlapping differentially expressed genes involved in regulation of cell division, we continued further investigations of the peculiar features of cell cycle of strains carrying *sup35-218* nonsense allele. To evaluate the effect of the *sup35-218* nonsense allele on the cell cycle progression, we compared the distribution of DNA content between unsynchronised cultures of wild-type and *sup35-218* strains by flow cytometry (Figure 6).

We observed two similar clear peaks of fluorescence intensity in wild-type strains, which corresponds to the DNA content of cells with unreplicated (1C) and replicated DNA (2C) in unsynchronised cultures. In contrast, cells with the *sup35-218* allele mostly did not demonstrate clear peaks of fluorescence intensity that would have corresponded to 1C or 2C. Additionally, the comparison of independent clones bearing *sup35-218* with each other did not reveal any patterns in the distribution of DNA amount. Therefore, cells containing the *sup35-218* nonsense allele were characterized by a greater heterogeneity in DNA content compared to the wild-type, which may indicate the disruption of cell cycle progression in cells with *sup35-218* nonsense allele.

## 3. Discussion

In this study, we describe global changes in the gene expression profile of a yeast cell in response to the introduction of a nonsense mutation in the essential *SUP35* gene encoding a release factor, eRF3. We discovered 1366 differentially expressed genes between cells bearing a wild-type *SUP35* allele and a mutant *sup35-218* allele (Figure 1). In-depth functional analysis of this set of DEGs suggests that several biological processes are involved in adaptation to such pronounced translation termination defects, with metabolic processes and cell cycle among the major discoveries. While massive metabolic changes are expected during stress (e.g., [20,21]), the observed upregulation of certain cell cycle components attracts attention, as it might explain gene amplification observed previously in strains bearing *sup35-218* [10].

Normally, the yeast cell cycle can be divided into four phases. During the first of these phases, the growth phase (G1), there is a significant cell growth and increase in cell volume. This stage is followed by the synthetic (S) phase, during which the DNA replicates. After that, the cell enters the second phase of growth (G2), in which a bud appears; and the process ends with the mitotic phase (M), in which division occurs, giving rise to a daughter cell. *S. cerevisiae* cells, compared to fission yeasts, usually have a prolonged G1 phase; however, it depends on conditions, nutrients and size of the cell [22]. Cell cycle progression is mainly controlled by proteins of the cyclin-dependent kinase (CDK—Cdc28 in yeast) family and their interaction with cyclins whose expressions are fluctuating during the cell cycle [23]. Importantly, cell cycle is regulated at the transcriptional, translational, and post-translational levels. In our work, we tried to use both transcriptomic and proteomic data to obtain more insights into the cell cycle changes observed in cells carrying *sup35-218*; however, the datasets showed a low degree of overlap between sets of differentially expressed genes and proteins, making it harder to draw an unambiguous conclusions regarding exact state of cell cycle regulation in mutant cells.

The expression of the central coordinator of the major events of the yeast cell division cycle, *CDC28*, was found to be increased at both transcriptomic and proteomic level. Cdc28 activity controls the timing of mitotic commitment, bud initiation, DNA replication, spindle formation, and chromosome separation [24]. While the abundance of Cdc28 was substantially altered in strains carrying *sup35-218*, its impact on the phenotype could not be determined, in particular, due to additional regulation of Cdc28 at the post-translation level (for example, via post-translational modifications) [25].

Additionally, we observed an increase in the transcription level of *FKH1*, which encodes the Forkhead homolog protein and its forkhead-associated (FHA) domain acts in promoting the ORC-origin binding and origin activity at a subset of origins in *S. cerevisiae* [26]. Previously it was shown that an increased expression of the FKH transcription factors leads to an extended lifespan, improved stress response and rescues APC mutant growth defects [27].

Besides *CDC28* and *FKH1*, we discovered increased expression of the *ORC2*, *SWI6*, and *MCM2*, primarily involved in DNA synthesis, and genes encoding cyclins of the CLN and CLB families (*CLN3*, *CLN2*, *CLB2*, *CLB5*, and *CLB6*). Cyclins are regulated both at the transcriptional and post-transcriptional level, and they are expressed in alternate phases of the cell cycle. The CLN family consists of Cln1, Cln2, and Cln3, which are expressed mostly in the G1 phase. The CLB-type cyclins, on the other hand, regulate later cell cycle stages, including DNA replication and the entry into mitosis. To prevent premature DNA replication, the activity of the first B-type cyclins to be expressed, Clb5/6, is inhibited through high levels of the cyclin-dependent kinase inhibitor Sic1 during G1 [28]. Notably, we did not identify cyclins as differentially abundant in the proteome; however, this finding is rather expected given the low abundance of these proteins.

In addition to changes in cyclin/CDK abundance, downregulation of the proteins that comprise the APC/C complex or interact with it (*APC9*, *CDC23*, and *CDC20*) was detected. The APC/C complex is one of the crucial regulators of mitosis progression. This complex has two main forms—APC/C-Cdc20 and APC/C-Cdh1. These two forms have overlapping as well as distinct substrates. In *S. cerevisiae*, APC/C-Cdc20 is required for the degradation of Pds1 (securin) and the B-type cyclin Clb5, whereas APC/C-Cdh1 promotes the degradation of Clb2 [29,30].

Apc9 is a nonessential component of the APC; however, deletion mutants of *APC9* have delayed progression through mitosis [25]. Cdc23 was identified as a conserved subunit of APC/C [31], and it was shown that *cdc23* mutants are defective in nuclear division in *S. cerevisiae* [32]. Previous studies showed that *S. cerevisiae* strains bearing a *cdc23* mutation represented a metaphase-like arrest phenotype. In addition, yeast cells with mutation in *CDC23* also showed defects in both entering and exiting anaphase [33], suggesting that Cdc23 may play an important role in at least two stages of the cell cycle, metaphase-to-anaphase transition and telophase-to-G1 transition [34,35,36,37].

*CDC20* expression is regulated at both transcriptional and post-transcriptional levels [38]. Mice embryos lacking the Cdc20 function demonstrated the metaphase arrest at the two-cell stage with high levels of cyclin B1, indicating an essential role of Cdc20 in mitosis that is not redundant with that of Cdh1 (Cdc20 homolog, activator of anaphase-promoting complex) [39]. In fission yeast *Schizosaccharomyces pombe*, the transcriptional silencing of Cdc20, an APC activator, was shown to cause cell cycle arrest in metaphase accompanied by high levels of cyclin B and securin Pds1, which inhibits the separase Esp1. Most cells that remain in mitosis for a long time undergo apoptosis, but some of them skip cytokinesis and enter G1 with non-segregated chromosomes. This process, called mitotic slippage, increases genome instability and results in aneuploidy [40]. If a similar process can occur in *S. cerevisiae* cells upon *CDC20* downregulation, such slippage can provide a mechanistic basis for the observed amplification of *sup35-n* and *sup45-n* alleles [10].

Taken together, elevated levels of cyclins (at G1, S, and G2 stages) and the Cdc28 kinase, as well as a decrease in the expression of the anaphase-stimulating complex components, allows us to suggest that amplification of plasmids or chromosomes detected in our previous work could occur due to a slowdown in the cell cycle (Figure 7). First, defects and delays in the synthetic phase could allow for intensive replication of plasmids. At the same time, a decrease in the efficiency of the APC/C and Cdc20 complex, in turn, could lead to disturbances in chromosome segregation and plasmid segregation. In earlier studies, it has been shown that mutation in *SUP35* gene influences the cell cycle and leads to the accumulation of large buds, disruptions in DNA synthesis and G1-to-S phase transition arrest [41]. Such mutants were not capable of either continuing the cell cycle or copulating [42]. In 2002, Valouev et al., demonstrated the effect of eRF1 and eRF3 depletion on the yeast cell morphology and cell cycle progression [43]. Our results of flow cytometric analysis, which showed an accumulation of cells with non-1C DNA content in cultures harboring *sup35-218* mutation (Figure 6), differ from those obtained by Valouev et al. in eRF3-depleted cells. However, both our results and previously published data indicate that alterations in eRF3 abundance and/or activity lead to profound changes in DNA content, supporting the role of observed changes in expression of cell cycle regulators in mutant allele amplification.

It is important to note, however, that cell cycle disturbances are usually studied using synchronized cultures, despite the facts that synchronization affects cell cycle progression heavily and that single cell behavior deviates from population behavior [44]. As thus, further studies using synchronized cultures could help to completely disentangle the complex changes in cell cycle regulation and their impact on gene amplification and adaptation.

## 4. Materials and Methods

### 4.1. Yeast Strains and Media

The U-14-D1690 [10] strain was used in this study. This strain is isogenic to 1A-D1628 [8,45], and contains a pRSU2 [46] plasmid as the sole source of *SUP35*. During plasmid shuffling, we used the pRSU1 [46] and pRSU1-218 [9] plasmids, which are both derivatives of the centromeric pRS315 [47] vector and carried the *LEU2* selective marker. Strains obtained after the plasmid shuffling are designated L-14-D1690 and 218L-14-D1690, respectively.

Standard methods of cultivation and manipulation of yeast were used throughout this work. Yeast strains were cultivated at 26 °C in standard solid and liquid synthetic media (SC). The SC medium contained higher amounts of certain nutrients: 40 mg/L adenine, 20 mg/L L-histidine, 20 mg/L L-lysine, 20 mg/L L-methionine, 20 mg/L L-threonine, 20 mg/L L-tryptophan, 20 mg/L uracil, and 20 mg/L L-leucine. For strains bearing two plasmids (with *URA3* and *LEU2* marker genes), SC-UL media was used that did not contain either uracil or L-leucine. SC medium containing 1000 mg/L 5-fluoroorotic acid (5-FOA) (Thermo Scientific, Waltham, MA, USA) was used for selection against cells bearing plasmids with the *URA3* marker [48]. Yeast strains were incubated on 5-FOA medium for 4–5 days. The yeast transformation was carried out according to the standard protocol [49]. For flow cytometry, yeast strains were grown at 26 °C in YEPD medium supplemented with 40 mg/L adenine.

### 4.2. Cell Cycle Analysis by Flow Cytometry

For cell cycle analysis by flow cytometry, overnight cultures of individual clones were diluted into fresh YEPD medium until OD_600_ = 0.1, and incubated to pass 1–2 cell divisions (OD_600_ = 0.4). The sample preparation was conducted according to [50] with minor modifications: the incubation time of the RNAse solution (50 mM Tris pH8, 15 mM NaCl; 4 μg/mL RNAseA (Merck, Burlington, MA, USA)) was 2–3 h. The samples were sonicated 4 times for 5 s on ice. After adding 10 μL SYBR Green I solution (#PB025, Evrogen, Moscow, Russia), the samples were incubated for 0.5–1 h in the dark. The analyses were performed on the CytoFlex S (Beckman Coulter, Brea, CA, USA) at 488 nm through a 525/40 filter, collecting 50,000 events per sample. The FlowJo10 (BD Biosciences, Franklin Lakes, NJ, USA) was used for data analysis.

### 4.3. RNA-Seq Library Preparation and Sequencing

For extraction of the RNA, cultures were grown in 30 mL of SC-U or SC-L liquid medium until OD_600_ = 0.8–1.0. The cells were then harvested by centrifugation at 8000× *g* for 5 min and washed. The total yeast RNA was isolated using the GeneJET RNA Purification Kit #K0731 (Thermo Scientific, Waltham, MA, USA) according to the manufacturer’s instructions. The RNA concentration and quality were evaluated using a NanoDrop Spectrophotometer (Thermo Scientific, Waltham, MA, USA). Preparation of the libraries for sequencing was carried out using NEBNext® Ultra™ II Directional RNA Library Prep Kit for Illumina (#E7765, NEB, Ipswich, MA, USA) and NEBNext® Multiplex Oligos for Illumina® (96 Unique Dual Index Primer Pairs Set 2) (#E6442S, NEB, Ipswich, MA, USA). Sequencing was performed using the Illumina HiSeq 4000 platform in the paired-end mode and a read length of 150 nucleotides.

A total of 10 samples were used for sequencing (three biological replicates for the initial U-14-D1690 strain, four replicates for cells bearing the wild-type *SUP35* allele after plasmid shuffling, and three replicates for cells bearing the *sup35-218* allele).

### 4.4. RNA-Seq Data Analysis

Raw RNA-seq reads were aligned onto the yeast S288C (R64-3-1) genome using Hisat2 [51]. Quantification of gene expression was performed using feature counts [52]. Gene annotation was obtained from the Saccharomyces Genome Database (the same R64-3-1 version was used). Further analysis of the obtained gene count matrix was performed using the R v4.1.1 software [53].

To perform the principal component analysis of the gene expression profile, the original count matrix was transformed using the rlog function in the DESeq2 package (v1.34.0) [54]. Differential expression analysis was performed using the default DESeq2 functions. Differentially expressed genes were defined as genes with FDR-adjusted *p* < 0.05 and log2FC > 0.5 (1.4-fold increase in expression) or log2FC < −0.5 (1.4-fold decrease in expression) for up- and downregulated genes, respectively.

A Gene Ontology (GO) term enrichment analysis for the obtained DEGs was performed using the clusterProfiler package for R. For the analysis of the role of DEGs in the identified pathways, we used the Kyoto Encyclopedia of Genes and Genomes (KEGG) [55].

For the analysis of transcription factors that play a role in the observed transcriptional changes, the set of DEGs was analyzed using information system YEASTRACT (www.yeastract.com, accessed on 20 November 2023).

In order to calculate the correlation between the transcript and the protein levels, RNA-seq read counts were normalized to account for the gene length and library size, and logarithms were performed to transform the data. The Kendell correlation coefficient was then calculated using the built-in function in R v4.1.1 software [53].

### 4.5. RNA Extraction and cDNA Generation for qPCR

For extraction of RNA, cultures were grown in SC-L liquid medium until OD_600_ of 0.8–1, then the cells were harvested and washed. The total yeast RNA was isolated using the GeneJET RNA Purification Kit (Thermo Scientific, Waltham, MA, USA, #K0731) and treated with DNase I (RapidOut DNA Removal Kit, Thermo Scientific, Waltham, MA, USA, #K2981) according to the manufacturer’s instructions. The RNA concentration and quality were evaluated using a NanoDrop Spectrophotometer (Thermo Scientific, Waltham, MA, USA). Purified RNA was reverse transcribed with RevertAid RT Reverse Transcription Kit (Thermo Scientific, Waltham, MA, USA, #K1691). cDNA generation was performed under the following conditions: 25 °C for 5 min, 42 °C for 60 min and termination at 70 °C for 5 min.

### 4.6. qPCR

The expression levels of target cell cycle genes were analyzed by quantitative PCR (qPCR) with EVA Green 2.5X PCR-mix (Syntol, Moscow, Russia) according to the manufacturer’s instructions. The reactions and quantification were performed using CFX96 amplifiers (Bio-Rad, Hercules, CA, USA). The quantitation cycle for *ACT1* was used as a reference. Triplicate qPCRs were performed for each biological replicate. We analyzed the specificity of the amplification through the melting curves for all pairs of primers used for controls and genes of interest and, in all cases observed, a single peak accounting for a single PCR product. The ΔΔC_T_ method [56] was used to measure the levels of transcription. The resulting values were used to quantify the gene expression changes in cells containing the *sup35-218* allele. Primer pairs used in this analysis are listed in Table 2.

### 4.7. Protein Extraction

Cells were grown and pelleted as for RNA extraction. The total proteome was isolated using the following methodology. Precipitated cells were resuspended in lysis buffer (30 mM Tris-HCl, pH 7.4; 150 mM NaCl; 10 mM PMSF) [57] with a cocktail of protease inhibitors (Protease Inhibitor Cocktail, Sigma P8215). Next, glass beads were added to the suspension. Cell lysis was performed with Fast-Prep-24 homogeniser (MP Biomedicals) at a speed of 6.0 M/S, five times for 20 s each, with cells cooled in ice for 3 min between rounds. In the final step, the lysate was centrifuged for 10 min, 400× *g* at 4 °C. The protein concentration was measured with a Lumiprobe QuDye Protein kit (#15102, Lumiprobe, Hunt Valley, MD, USA). The average protein concentration was 5 µg/µL. The samples were then equilibrated to the minimum concentration and 30 µg of protein was taken for analysis. The obtained samples were treated with chemically pure trypsin (Trypsin Gold, Mass Spectrometry Grade (Promega #V5280, Madison, WI, USA)) overnight. Then, the peptides were purified according to the stage tips protocol [58]. The obtained samples were further used for mass spectrometric analysis.

### 4.8. Proteome Analysis

The analysis of the prepared proteomic libraries was performed using mass spectrometer timsTOF Pro 2 (Bruker, Billerica, MA, USA). To analyze the obtained data, we used PEAKS Studio 11 software, and the proteome of the *S. cerevisiae* strain (strain S288C, Uniprot ID—UP000002311) was used as a reference database. Protein counts were analyzed with the limma package (v 3.56.2) with R v4.1.1 software [53]. Differentially produced proteins were defined as proteins with adjusted *p*-value < 0.05 and log2FC > 0.5 (1.4-fold increase in abundance) or log2FC < −0.5 (1.4-fold decrease in abundance) for up and down regulated genes, respectively.

### 4.9. Code Availability

All code pertinent to the bioinformatic or statistical analysis presented in this work is available at https://github.com/mrbarbitoff/sup35n_expression_analysis, accessed on 4 June 2024. Raw and processed RNA-seq data have been submitted to the Gene Expression Omnibus (GEO) database (accession number GSE267888). Raw mass-spectrometry data have been uploaded to the Proteomics Identification (PRIDE) database (accession number PXD052727).

## Figures and Tables

**Figure 1 ijms-25-06308-f001:**
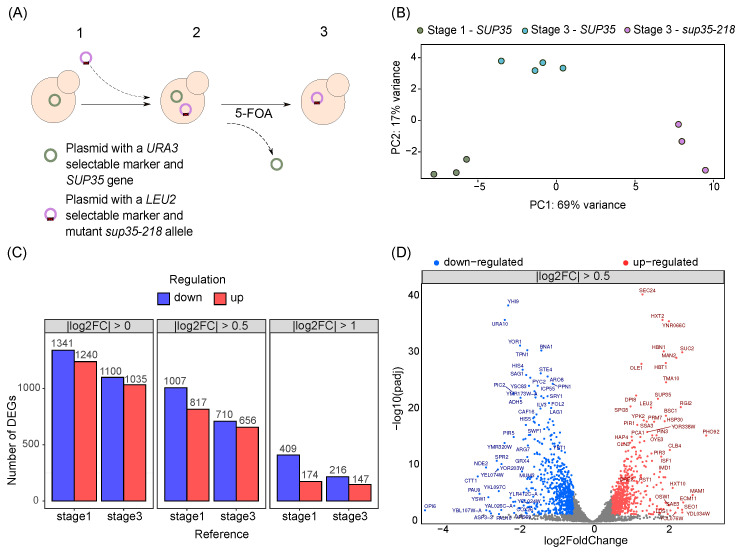
Transcriptome profile of strains harboring nonsense-mutation *sup35-218*. (**A**) Scheme illustrating the experimental design used to study yeast adaptation to nonsense mutations in release factor genes. (**B**) Scatterplot representing principal component analysis of 10 yeast transcriptomes carrying different combinations of alleles of the *SUP35* gene. (**C**) Comparative analysis of the number of differentially expressed genes, depending on the log2FC cutoff and set of controls used. (**D**) Analysis of differential gene expression in cells at the 3rd stage of the experiment, carrying the *sup35-218* allele compared to 3rd stage harboring the *SUP35* wild-type allele. Significantly differentially expressed genes (FDR < 0.05) are represented by red dots (upregulated) and blue dots (downregulated). Grey dots represent genes with unaltered expression.

**Figure 2 ijms-25-06308-f002:**
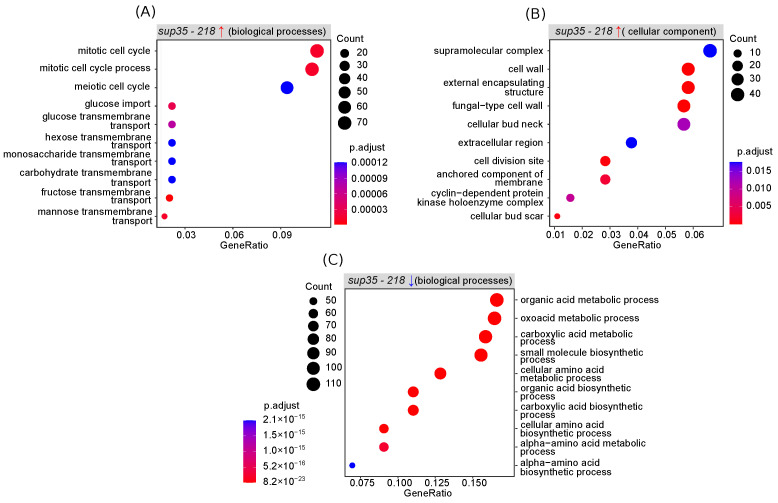
Results of gene set enrichment analysis using Biological Process (BP) and Cellular Component (CC) terms from Gene Ontology. The analysis was performed using the clusterProfiler package for R. The dot size is proportional to the enrichment ratio, and the color gradient represents the adjusted significance level. (**A**) Genes with increased expression in cells bearing *sup35-218* (biological processes). (**B**) Genes with increased expression in cells bearing *sup35-218* (cellular component). (**C**) Genes with decreased expression in cells bearing (biological processes).

**Figure 3 ijms-25-06308-f003:**
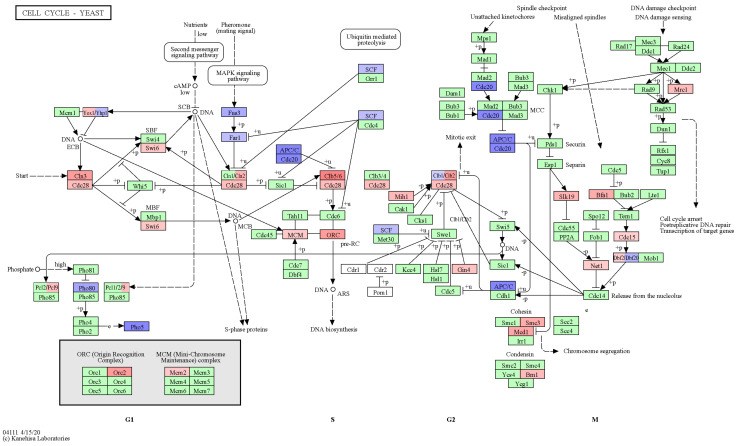
A diagram showing the changes in the expression of genes controlling the yeast cell cycle progression. Visualization of the relationship of genes involved in cell cycle control using KEGG PATHWAY. Genes with increased expression in the presence of the *sup35-218* allele are marked in red; genes with reduced expression are marked in blue. Color contrast reflects the degree of increase or decrease in expression. Cases where two genes with similar functions have inconsistent expression changes are highlighted with two colors (Yox1/Yhp1, Clb1/2, Dbf2/20). Genes where no significant changes in expression were found are highlighted in green.

**Figure 4 ijms-25-06308-f004:**
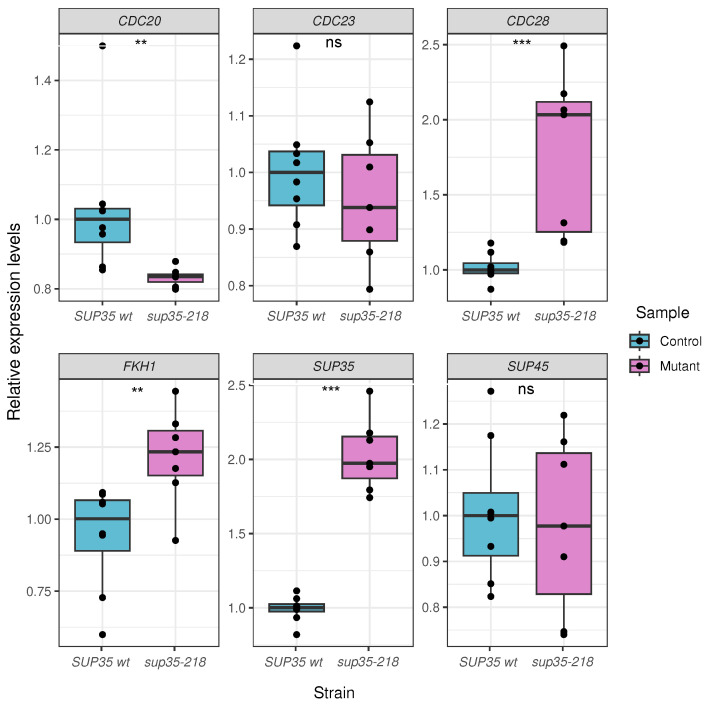
Validation of changes in expression of DEGs identified in RNA-Seq analysis using qPCR. Shown are boxplots of the relative expression levels calculated using the ΔΔCt method (see Section 4 for more details). ns—not significant, ** *p*-value < 0.05, *** *p*-value < 0.001 according to Wilcoxon–Mann–Whitney rank sum test. At least seven biological replicates were used for each group.

**Figure 5 ijms-25-06308-f005:**
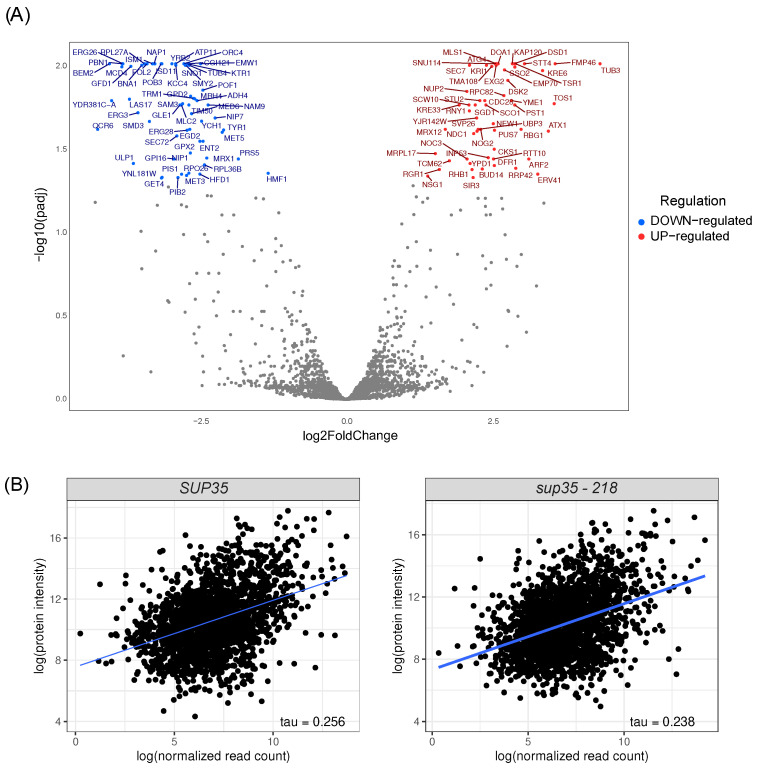
Proteome profile of strains harboring nonsense-mutation *sup35-218*. (**A**) Analysis of differential protein production in cells at the 3rd stage of the experiment, carrying the *sup35-218* allele compared to cells harboring *SUP35* wild-type allele. Differentially produced proteins (FDR < 0.05) are represented by red dots (upregulated) and blue dots (downregulated). Grey dots represent proteins with unaltered production. (**B**) Evaluation of the correlation of gene and protein expression levels in strains carrying the wild-type *SUP35* allele or mutant *sup35-218* allele (left and right plots, respectively).

**Figure 6 ijms-25-06308-f006:**
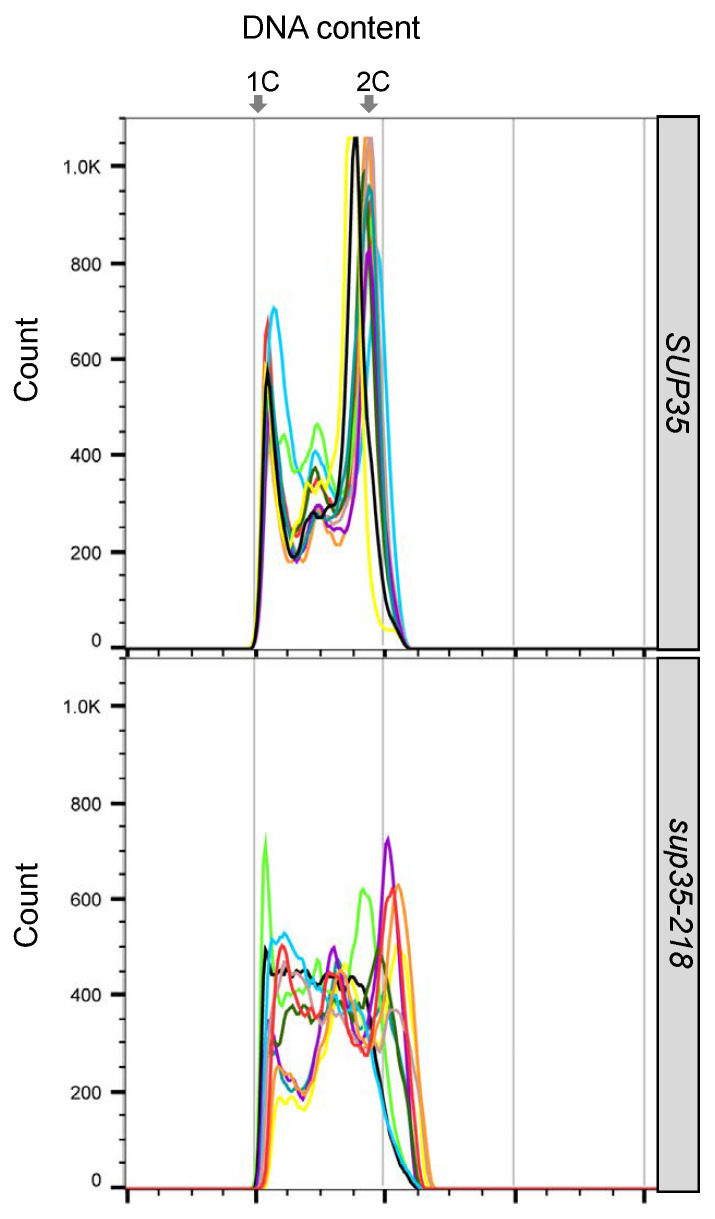
Cell cycle analysis by flow cytometry, illustrating the DNA content of yeast cultures of wild-type strain (L-14-D1690) or *sup35-218* mutant cells (218L-14-D1690). Ten independent yeast cultures were grown and then prepared for flow cytometry as described in Section 4. Curves representing independent clones are shown in different colors. Locations of peaks of the 1C and 2C cell populations are indicated by arrowheads.

**Figure 7 ijms-25-06308-f007:**
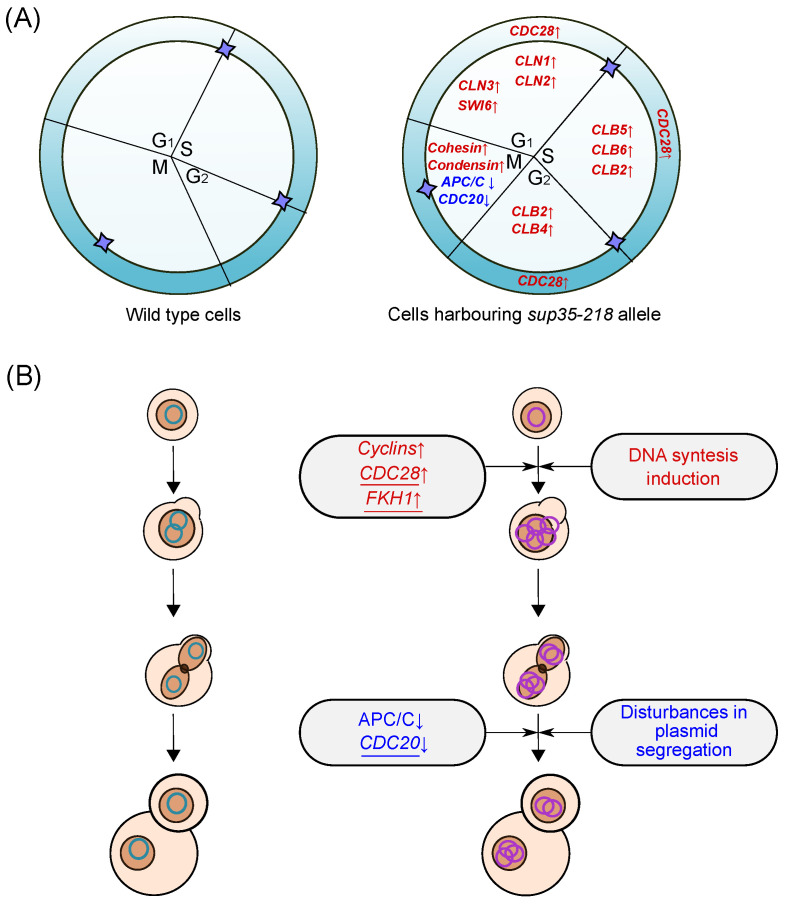
A schematic representation of the possible changes in the cell cycle in cells carrying the *sup35-218* mutation. (**A**) Changes in expression of genes responsible for the progression of cell cycle phases in cells carrying the *sup35-218* allele (right) compared to cells carrying the *SUP35* allele (left). Purple asterisks indicate checkpoints in cell cycle regulation. Genes with increased expression are depicted in red (as in Figure 3) and are marked with an up arrow; genes with decreased expression are depicted in blue and are marked with a down arrow. (**B**) Scheme of changes occurring during plasmid replication and segregation in cells carrying the *sup35-218* allele (right) compared to cells carrying the *SUP35* allele (left). Plasmids with wild-type *SUP35* are shown in green, plasmids with *sup35-218* allele are shown in blue. Genes with changes in expression validated using qPCR are underlined.

**Table 1 ijms-25-06308-t001:** Genes with concordant changes in expression level in the transcriptome and proteome.

Regulation	Gene	Protein	Description
UP	*YPR190C*	Rpc82	RNA polymerase III subunit C82
UP	*YLR045C*	Stu2	required for elongation of mitotic spindle during anaphase
UP	*YBR160W*	Cdc28	master regulator of mitotic and meiotic cell cycles
UP	*YDR055W*	Pst1	cell wall protein, up-regulated by activation of the cell integrity pathway
DOWN	*YGR267C*	Fol2	catalyzes first step in folic acid biosynthesis
DOWN	*YGL001C*	Erg26	take part in ergosterol biosynthesis
DOWN	*YJR025C*	Bna1	required for the de novo biosynthesis of NAD from tryptophan to kynurenine
DOWN	*YER044C*	Erg28	Endoplasmic reticulum membrane protein
DOWN	*YER057C*	Hmf1	takes part in maintenance of the mitochondrial genome

**Table 2 ijms-25-06308-t002:** Primers used for qPCR analysis in this work.

Primer Name	Target Gene	Sequence 5’→3’
ACT1-F	*ACT1*	TAACGGTTCTGGTATGTGTAAAGC
ACT1-R	*ACT1*	GCTTCATCACCAACGTAGGAGTC
SUP35-F	*SUP35*	ACAACAAGGTAACAACAGATACC
SUP35-R	*SUP35*	GGATTGAATTGCTGCTGATAAC
SUP45-F	*SUP45*	CGATCCAAGACTAGCATGTAAG
SUP45-R	*SUP45*	CTTGAACATACTTGACATTGGC
FKH1-RT-F	*FKH1*	CGGATCATGGGAATTACAG
FKH1-RT_R	*FKH1*	CACACCACCTATGTCTATG
CDC20-RT-F	*CDC20*	GTCGATCCTGAAACCTTAC
CDC20-RT-R	*CDC20*	CATATCTAACCCACACGC
CDC23-RT-F2	*CDC23*	AGATGTCACTAATCCATACCT
CDC23-RT-R2	*CDC23*	CATTTCGTCCGTAAACTTCC
CDC28-RT-F	*CDC28*	AGAGTAGTCGCATTGAAGAAA
CDC28-RT-R	*CDC28*	GCATCAGAGTGAACAATATCG

## Data Availability

Code used for data analysis, as well as intermediate data files, are publicly available at https://github.com/mrbarbitoff/sup35n_expression_analysis, accessed on 4 June 2024. Raw and processed RNA-seq data have been submitted to the Gene Expression Omnibus (GEO) database. Raw mass-spectrometry data have been uploaded to the Proteomics Identification database (PRIDE).

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
