# Peer review of "Gene Expression Analysis of Yeast Strains with a Nonsense Mutation in the eRF3-Coding Gene Highlights Possible Mechanisms of Adaptation"

_ijms, 2024, doi:10.3390/ijms25126308_

Round 1
Reviewer 1 Report (Previous Reviewer 2)
Comments and Suggestions for Authors
The manuscript was well revised including qRT-PCR analysis, but there were still two comments, 1, authors have accepted “log2FC > 0.5”, which means 1.414214 folds for different (up) expressing or 0.7071068 for down-expressing, please confirm it. 2, authors have submitted data to GEO database, but missed the accession number, please add it in “Data Availability Statement” section. Good luck.
Author Response
The manuscript was well revised including qRT-PCR analysis, but there were still two comments, 1, authors have accepted “log2FC > 0.5”, which means 1.414214 folds for different (up) expressing or 0.7071068 for down-expressing, please confirm it. 2, authors have submitted data to GEO database, but missed the accession number, please add it in “Data Availability Statement” section. Good luck.
Authors: We thank the Reviewer for the positive assessment of our work. We have tried to correct all of the issues mentioned in the Review.
- We confirm that the fold changes given by the Reviewer are correct. We have added a simpler definition of log2FC cutoffs in terms of the fold change in expression/abundance to the RNA-seq data analysis section and Proteome analysis section.
- The GSE identifier of the RNA-seq dataset has been added (when submitting the article, it was not specified due to ongoing processing of data at GEO).
Reviewer 2 Report (Previous Reviewer 1)
Comments and Suggestions for Authors
Gene expression analysis of yeast strains with a nonsense mutation in the eRF3-coding gene can provide valuable insights into the cellular mechanisms of adaptation. The eRF3 gene, also known as SUP35, plays a crucial role in the translation termination process by interacting with the release factor eRF1 to promote the hydrolysis of the peptidyl-tRNA bond, thereby releasing the nascent polypeptide from the ribosome. It looks sound.
Author Response
Gene expression analysis of yeast strains with a nonsense mutation in the eRF3-coding gene can provide valuable insights into the cellular mechanisms of adaptation. The eRF3 gene, also known as SUP35, plays a crucial role in the translation termination process by interacting with the release factor eRF1 to promote the hydrolysis of the peptidyl-tRNA bond, thereby releasing the nascent polypeptide from the ribosome. It looks sound.
Authors: Thank you for the positive assessment of our work!
This manuscript is a resubmission of an earlier submission. The following is a list of the peer review reports and author responses from that submission.
Round 1
Reviewer 1 Report
Comments and Suggestions for Authors
This article is related to the author's previous papers [PMID: 34946968]. Although the author provides a correlation between the cell cycle and eRF3, this is a known biological function of eRF3 and is not novel enough for publication.
It is necessary to explain whether eRF3 is related to drive some genes or pathways by their data. This paper lacks novelty due to the previous publications.
Reviewer 2 Report
Comments and Suggestions for Authors
In the manuscript named “Gene expression analysis of yeast strains with a nonsense mutation in the eRF3-coding gene highlights possible mechanisms of adaptation”, Evgeniia M. Maksiutenko et al have performed RNA-seq analysis and proteome analysis to reveal gene expression changes during cellular adaptation process, and their results have demonstrated significant changes in the transcription of genes that control the cell cycle. These results were helpful for determining yeast development, and exploring molecular mechanism of cell cycle progression. The manuscript was well prepared, but there are some comments about this manuscript.
1. The “mg/l” and “g/L” should be unified, see line 72-78.
2. Why is the DEG identified with “log2FC > 0.5”, in line 112, which is always set as “1” in other articles. In addition, in proteome analysis the parameter was set as “log2FC > 2”, line 143. Please explain.
3. There is no molecular experiment to validate their high-throughput sequencing results, for example, qRT-PCR, could you supply it?
4. Raw RNA-seq and proteome data should be submitted to public database, not upon request.
Reviewer 3 Report
Comments and Suggestions for Authors
The article titled "Gene expression analysis of yeast strains with a nonsense mutation in the eRF3-coding gene highlights possible mechanisms of adaptation" presents an investigation into cellular adaptation mechanisms in Saccharomyces cerevisiae in response to nonsense mutations in translation termination factor genes (eRF1 and eRF3). The introduction effectively establishes the background context, elucidating the significance of protein synthesis termination, nonsense mutations, and potential therapeutic implications. Additionally, the rationale for employing Saccharomyces cerevisiae as a model organism is well-explained, along with referencing previously identified strains with nonsense mutations in SUP45 and SUP35 genes.
Major comments:
RNA-Seq Library Preparation and Sequencing, Data Analysis
However, it would be useful to mention the number of biological replicates used for RNA-Seq. Without appropriate statistical controls, the validity of the identified DEGs is questionable.
Additionally, the authors should provide more details on the normalization method used for gene expression quantification and specify the statistical test used for identifying differentially expressed genes in the RNA-seq data analysis section. The rationale behind choosing a log2 fold change threshold of 0.5 is unclear and seems arbitrary.
Protein Extraction Protocol:
While the lysis buffer composition is provided, essential details such as the volume of buffer per unit of cell pellet or the ratio of buffer to cell biomass are conspicuously absent. The authors should mention the volume of lysis buffer used for resuspending the cells and the volume and size of glass beads added to the suspension. Furthermore, the authors should provide more details on the limma package analysis, such as the normalization method and the statistical test used for identifying differentially produced proteins.
The broken code link (line 147) should be rectified to ensure accessibility to relevant resources.
Additional Recommendations:
It is a common practice for authors to provide supplemental data of processed results. Additionally, publicly available databases such as Gene Expression Omnibus (GEO) and ArrayExpress allow to share and access transcriptomics data from various studies, and similar resources exist for proteome data.
Validation of the RNA-seq data using qRT-PCR or alternative methods to confirm observed gene expression changes would enhance the reliability and credibility of the findings.
Conclusion:
In conclusion, while the study addresses an important research question, several methodological flaws and omissions undermine the integrity and interpretability of the results and should be corrected.